# Advancing Molecular Weight Determination of Lignin by Multi-Angle Light Scattering

**DOI:** 10.3390/polym16040477

**Published:** 2024-02-08

**Authors:** Mason L. Clobes, Evguenii I. Kozliak, Alena Kubátová

**Affiliations:** Department of Chemistry, University of North Dakota, 151 Cornell St., Stop 9024, Grand Forks, ND 58202, USA; mason.clobes@und.edu

**Keywords:** multi-angle light scattering, technical lignin, radius of gyration, second virial coefficient, differential refractive index increment, Rayleigh–Gans approximation

## Abstract

Due to the complexity and recalcitrance of lignin, its chemical characterization is a key factor preventing the valorization of this abundant material. Multi-angle light scattering (MALS) is becoming a sought-after technique for absolute molecular weight (MW) determination of polymers and proteins. Lignin is a suitable candidate for MW determination via MALS, yet further investigation is required to confirm its absolute MW values and molecular size. Studies aiming to break down lignin into a variety of renewable products will benefit greatly from a simple and reliable determination method like MALS. Recent pioneering studies, discussed in this review, addressed several key challenges in lignin’s MW characterization. Nevertheless, some lignin-specific issues still need to be considered for in-depth characterization. This study explores how MALS instrumentation manages the complexities of determining lignin’s MW, e.g., with simultaneous fractionation and fluorescence interference mitigation. Additionally, we rationalize the importance of a more detailed light scattering analysis for lignin characterization, including aspects like the second virial coefficient and radius of gyration.

## 1. Introduction

As the world is trending towards limiting fossil fuel consumption, new, abundant raw materials need to be investigated for their ability to replace petroleum-based goods. One resource to consider is plant biomass, whose three main components are cellulose, hemicellulose, and lignin. The latter is estimated to constitute nearly 30 wt % of plant biomass, making it one of the most abundant bio-polymers [1,2,3]. Outside of its natural occurrence in plants, chemical pulping industries extract up to 70 million metric tons of lignin annually as a waste product [4,5,6]. Both the paper and agricultural crop processing industries utilize carbohydrates from plant biomass while lignin is removed and consumed as a low-grade fuel or disposed of by other methods [5,6].

Lignin’s polymeric structure consists of three major monomers, *p*-coumaryl, coniferyl, and sinapyl alcohols called monolignols [3,5]. These monomers are connected with various linkages, creating a large aromatic heteropolymer. Variation in lignin arises naturally between plant species, but different isolation processes further increase the variety by modifying its structural makeup [5,7]. After the isolation of lignin from biomass, the products are referred to as “technical lignins.” Specific isolation treatments result in different types of technical lignins. The kraft isolation process (NaOH/Na_2_S) [8] is commonly used in the paper pulping industry, resulting in a “kraft” lignin. Treatments involving sulfurous acid yield lignosulfonates, while extraction with hot organic solvents such as acetone, methanol, or acetic acid produces “organosolv” lignins [9].

MW determination is essential for the development of lignin valorization methods via depolymerization. First of all, MW can serve as a direct indicator of degradation, as lowering the MW signifies the breakdown of a polymer. Modern methods of lignin degradation utilize either high temperature or rather reactive agents, such as photosensitizers, oxidants, enzymes, or metal catalysts [6,10,11,12,13,14,15,16]. As a result, the targeted depolymerization is accompanied by unwanted cross-linking side reactions, a phenomenon called re-polymerization. For example, when using so-called lignin model compounds, i.e., phenolic dimers or oligomers containing a specific characteristic bond to be cleaved, the yield of monomer phenolics is near-quantitative [17,18,19,20,21,22]. However, when heteropolymeric lignin is used as a feedstock instead of model compounds, the targeted products’ yields are usually much lower [23,24,25]. Even when the same protocol is applied, most target products fall below 30 wt % of the initial feedstock [26]. The other main products include high-MW coke and tar, so-called bio-oil, and relatively undefined oligomers [27].

Therefore, to monitor the process of lignin depolymerization, it is important to monitor not only the average mass of lignin and its degradation products but also its distribution over fractions of narrower MW ranges. This is why size exclusion chromatography (SEC) is frequently used and new methods, e.g., asymmetrical flow field-flow fractionation (AF4), are being improved upon to provide consistent fractionation for lignin analysis [1,4,28]. Hyphenation of these separation methods to multi-angle light scattering (MALS) detectors is an effective way of determining the absolute MW of fractionated, polydisperse samples such as lignin.

### 1.1. Current Methods of Lignin’s MW Determination

The determination of absolute molar mass appears to be the first step toward comprehensive characterization of lignin and could simplify the search for lignin applications. MW is commonly estimated by SEC, which is based on the molecular sieve effect, assuming that other analyte-stationary phase (unwanted) interactions are minimized. As a result, smaller molecules elute at later times while larger species elute earlier. The unwanted interactions are considered insignificant when calibration standards match the chemical nature of the analyte used. However, this issue remains a challenge when working with complex materials such as lignin, as there is no explicit series of standards that would apply to lignin samples [1,29,30]. This discrepancy in standards can generate errors in MW determination.

Other less common methods of MW determination include mass spectrometry (MS) [29,31,32], analytical ultracentrifugation (AUC) [33], and nuclear magnetic resonance (NMR) spectroscopy [34,35,36]. AUC is limited to monodisperse systems and requires tedious calculations from sedimentation equilibrium experiments to determine the weight average MW [33]. NMR spectroscopy requires expensive equipment and is susceptible to errors due to high polydispersity in unfractionated lignin samples [35]. Nonetheless, it is being employed for fast analysis of lignin’s MW [34,35]. Mass spectrometry with appropriate ionization techniques (usually atmospheric pressure ionization) have reported MW values and structural information [37]. However, MS yields MW values that are inherently biased toward lower values, due to the hindered transfer of high MW species to the gas phase prior to, during, or after ionization [38].

The inherent complexity of lignin brings forth challenges in the application of many instrumental analysis techniques. The fact that MALS instrumentation is capable of maintaining a theoretical relationship to determine MW makes it a promising option for lignin analysis.

### 1.2. MALS for Lignin’s MW Determination

A thorough explanation of light scattering theory and a review of its application to MW determination is provided in the book of Podzimek [39,40,41,42]. The monography of Øgendal [43] provides a theoretical review of MALS in a less technical language. The theoretical relationship between light scattering and molar mass allows for MW determination without the use of standards or calibration curves. MALS can analyze a wide array of materials ranging from natural rubbers [44,45] to metallic nanoparticles [46,47,48] and even globular proteins [49]. Overall, MALS is most commonly used in polymer studies [50,51,52] but may advance lignin analysis [1,7,30,53,54,55,56,57]. Significant challenges accompanying lignin’s MW determination via MALS are limited analyte solubility and interference by fluorescence. In addition, as explained by Podzimek, monodisperse sample preparation is crucial for the determination of MW, root mean square (RMS) radius, and molar mass moments [41]. Thus, separation into monodisperse fractions is essential while ensuring full solubilization in a compatible solvent for both the separation system and the MALS instrument. Multicomponent solvent systems are commonly used to solubilize lignin, but solvent mixtures may interfere with RI and MALS measurements [58]. The natural fluorescence observed in many lignin samples has been investigated, and its mitigation resulted in more accurate MW values compared to previous data [7,53].

### 1.3. MALS Theory

The foundation of MALS is based on the theory of static light scattering (SLS), where the Rayleigh–Gans approximation relates the intensity of scattered light at specific angles to the Rayleigh ratio (light scattered vs. incidental light—R(θ)) reflected in Equation (1) [53,59].
(1)$$R\left(\theta \right)=K^{*}McP\left(\theta \right)[1-2A_{2}McP\left(\theta \right)]$$

The optical contrast constant ($K^{*}$) describes the refractive index of the analyte depending on the medium, while the scattering form factor (sometimes called particle scattering function or just form factor) (P(θ)) is the ratio of scattered light at a specific angle compared to the zero angle [39]. The particle shape influences its scattering function [39]. The scattering form factor is not a crucial variable in determining MW since it is equal to unity at the zero scattering angle [41]. The second virial coefficient ($A_{2}$) describes the molecules’ affinity for the solvent. Finally, the molar mass (M) can be determined when the concentration (c) of the analyte is known. The first term ($K^{*}McP\left(\theta \right)$) is the major contributor to the approximation. This, in turn, illustrates the priority that needs to be given to accurate calculations of the optical contrast constant (Equation (2)). The second term ($\left[1-2A_{2}McP\left(\theta \right)\right]$) is a correction factor on which absolute MW determinations and their accuracy depend. Non-lignin polymers, when assuming $A_{2}=0$, only yield a 2% error in MW determination compared to accurate $A_{2}$ value calculations [41]. Determining the constants within the first term of Equation (1) is required for calculating absolute MW. The optical contrast constant is defined in Equation (2) [53].
(2)$$K^{*}=\frac{4\pi^{2}n_{0}^{2}}{N_{a}\lambda_{0}^{4}}{\left(\frac{dn}{dc}\right)}^{2}$$

Besides Avogadro’s number ($N_{a}$), the experimental factors affecting the $K^{*}$ value include the refractive index of the solvent ($n_{0}$) and analyte solution (n), and the wavelength of incident light ($\lambda_{0}$). The key parameter of Equation (2), dn/dc (in mL g^−1^), is commonly referred to as the differential refractive index increment and relates the change in refractive index (RI) to the change in concentration (c) of the solution. The commonly practiced addition of an RI detector to the MALS setup provides the capability to determine dn/dc values. Based on Equations (1) and (2), a MALS-RI combination is capable of providing MW determinations of numerous materials, and the addition of fractionation enables precise MW determination of individual fractions separated by their hydrodynamic size [41].

Given the attractive features of MALS, lignin has become a target analyte in studies addressing MW determination [1,7,30,53,54,55,56,57]. These studies successfully grasp fluorescence and solubility challenges involved with MALS application to lignin. The sections below will focus on the major achievements in lignin’s MW determination via MALS while highlighting advantageous areas for further investigation such as virial coefficients and radius of gyration.

## 2. Differential Refractive Index Increment Determination

The role of dn/dc in MW determination is seen in Equations (1) and (2) correlating directly to the determined MW. Since the dn/dc value is based on a specific solvent’s RI, each solvent and analyte combination has a unique dn/dc value. Two common methods of determining dn/dc values have been established in the literature. A consistent approach, known as “batch” determination, uses manual injections directly into an RI detector to determine dn/dc values. The injections often consist of 4–6 different concentrations between 0.1 and 3.0 mg mL^−1^ [1,30,53,54,55]. The samples are injected from low to high concentration, and the refractive index values are plotted against the concentration (in g mL^−1^). The slope displayed by this plot corresponds to the dn/dc value for that specific solvent and analyte combination. Variability in lignin types and solvents used for analysis makes dn/dc determination necessary for each lignin of interest (Table 1). However, this process is labor-intensive and time-consuming. According to Podzimek, batch determination is susceptible to error as any flaw in concentration (e.g., due to impurities) will create error in dn/dc values, which is carried to the MW calculations [39].

A fast and practical method for determining dn/dc is known as “online” determination, which can be performed at the same time as the MW determination when a SEC–MALS combination is used. Assuming 100% sample elution out of the SEC column, a single injection into the SEC–MALS system is used to estimate the dn/dc value. The “online” determination is conducted by estimating concentrations at specific times throughout the elution profile based on RI measurements. Similar to the “batch” method, the concentrations and the corresponding refractive indices plotted against each other can provide an estimate of the dn/dc value. “Online” estimations of the dn/dc values of lignin have been shown in recent literature to be an established protocol [7,56]. If there are impurities in the sample or solvent, the “online” method will have one half the error compared to the “batch” method, but either error will be carried directly to the MW calculations [39].

Zinovyev et al. used an RI detector wavelength (658 nm) that did not match the wavelength of the light scattering detector (785 nm) [7]. The error created by the 658 nm laser was estimated by Zinovyev et al. with the Cauchy relation (Equation (3)),
(3)$$\frac{dn}{dc}=A+\frac{B}{\lambda^{2}}$$
where *A* and *B* are constants [7]. It was concluded that dn/dc values are smaller at higher wavelengths with only a 1–3% difference in MW based on Equation (3) for the wavelength of lasers used [7]. The error was concluded to be insignificant since it is represented in both the MW and the dn/dc of the experiment [7]. Other studies referenced in this review determined dn/dc values using the “batch” method [1,30,53,54,55,56], which is more susceptible to error than “online” determination [39]. Currently, there is no comprehensive set of lignin data that displays the variation of calculated MW due to dn/dc determination methods. A comparative study between “batch” and “online” dn/dc determination would be advantageous for future method development and provide support to new claims.

Comparing reported dn/dc values of technical lignins is difficult due to different setups, i.e., online vs. batch methods, differing wavelengths, variety in lignin feedstocks and solvents. Table 1 displays determined dn/dc values and how laser wavelengths, solvents, and determination methods alter the calculated values. The study by Gidh et al. was one of the earliest cases of lignin characterization by MALS [53]. This work confirmed that changing solvents does alter dn/dc values, reiterating the importance of predetermining the dn/dc value before calculating the MW. Both Zinovyev et al. and Gaugler et al. tested similar types of lignin, yet the results were not exactly comparable, as the wavelengths, solvents, and SEC methods differed from one another [7,54]. It cannot be concluded which experimental variables exerted the highest impact on the results, but, as shown in Equation (2), both solvent RI and incident wavelength are significant factors.

It was concluded by Ponnudurai et al. that the largest error in lignin fraction MW determination occurs when the heterogeneous polymer (feedstock) has a different dn/dc value than the narrowly distributed MW fractions (obtained by acetone/water isolation) [1]. If the dn/dc used to determine the fraction’s MW is that of the feedstock and varies from the lignin fraction’s dn/dc, MW determinations will be incorrect, resulting in high error [1]. The absolute MW value is determined when the dn/dc value for each lignin fraction is applied to the MW calculations [1]. This conclusion increases the workload further for batch method as characterization requires a unique dn/dc value to be determined for each MW fraction. “Online” dn/dc determination is advantageous in this scenario as each fraction eluting from the SEC column can yield an individual dn/dc value. A comparison of “online” vs. “batch” dn/dc calculations of individual lignin fractions could greatly reduce the sheer labor if “online” modes were favored. The absolute MW of lignin fractions falls between the underestimation of SEC and the overestimation of SEC–MALS when only considering the dn/dc value of the feedstock lignin [1]. Previously reported acetone-dissolved lignin fractions’ dn/dc values increase as MW increases [1]. By contrast, as the MW and dn/dc values increase, the number of phenolic hydroxyl groups decreases [1]. The phenolic hydroxyl group content was believed to be the most likely explanation for the variation in dn/dc values for each fraction [1].

In summary, when considering the range of the determined MWs of lignins, it is apparent that the differences in dn/dc values impact MW determinations. However, the difference can also be related to the extent of solubilization. Emphasis needs to be placed on determining the correct dn/dc values for every analyte of interest. Lignin-solvent fractions require RI measurements at each solvent condition. The other option would be re-dissolving each fraction in a single solvent before performing an analysis. Careful consideration needs to be taken when selecting a determination protocol, as both “batch” and “online” methods can present errors in MW values.

## 3. Radius of Gyration Measurements

Lignin, due to its complex structure combined with a relatively low MW, cannot be automatically assumed to occur in solutions as a near-spherical, random coil, unlike many other polymers [38]. Thus, the shape of lignin will not abide by the confinements of well-defined scatterers presented by Podzimek [39]. It is a challenge to define the scattering form factor, and determining the shape solely from light scattering is not advisable [60]. The combined determination of size and shape obtained through other microscopy-based techniques appears to be the most effective approach to physical structure elucidation [60]. Fortunately, MW determination is not hindered by an unknown form factor. The scattering form factor is equal to unity at the zero scattering angle, obtained by approximation, e.g., using the Zimm plot [41].

The scattering form factor (P(θ) in Equation (1)) expresses angular dependence of the intensity of scattered light (demonstrated in Equation (4)),
(4)$$P(\theta )=\frac{R_{\theta }}{R_{0}},$$
where the Rayleigh ratios are represented at angles *θ* and zero [61]. Particles whose sizes are much smaller than that of the incident light ($<\frac{\lambda }{30}$) will produce isotropic scattering, where scattering is equal in all directions [60]. Larger particles ($>\frac{\lambda }{30}$) result in anisotropic scattering, where their size is related to the scattering form factor (Equation (5)) [60].
(5)$$P\left(\theta \right)=1-\frac{16\pi^{2}n_{0}^{2}}{3\lambda_{0}^{2}}\sin^{2}\left(\frac{\theta }{2}\right)\langle R_{G}^{2}\rangle +\ldots$$

Equation (5) relates the scattering form factor to the radius of gyration ($R_{g}$), which is the average radial distance of a particle’s mass elements to its center of mass [60]. The angular dependence of scattered light at angle zero yields the radius of gyration [39]. Every mass element ($m_{i}$) of the particle is at a certain distance ($r_{i}$) from the center of mass, which can be derived from light scattering measurements (Equation (6)) [60].
(6)$$\langle R_{G}^{2}\rangle =\frac{\sum_{i}m_{i}r_{i}^{2}}{\sum_{i}m_{i}}$$

It is assumed that the particle’s mass elements are independent Rayleigh scatterers rigidly attached [60]. Fractionation methods such as SEC or AF4 are critical in providing monodisperse samples for accurate size determinations [60]. The Debye, Zimm, and Berry formalisms are often used to eliminate the unknown form factor by determining scattering results at the unmeasurable zero angle [41]. With modern instrumentation, accurate measurements of light scattering at small angles have resulted in errors of less than 10% in radii ranging in sizes and structures [60]. Thus, the radius of gyration is commonly derived from the extrapolation at low angles based on the angular dependence of scattering, which will be discussed further in the next section (Figure 1). The capability of determining the radius of gyration for lignin allows for the scattering form factor to be calculated from Equation (5). This is important for MW determinations of lignin beyond a first-order approximation of Equation (1). The correction factor, or second term in Equation (1), can be calculated after determining the second virial coefficient.

The explicit mark between an isotropic scatterer and an anisotropic scatterer is not strictly defined [43]. In the simplest scenario, a spherical particle, it is generally accepted that isotropic scattering occurs when the particle size is 1/30 of the wavelength of light [60]. One of the experts in the field of light scattering, Philip J. Wyatt, concluded that the smallest spherical particle size measurable by a MALS instrument was 10 nm [60]. MALS lasers for lignin applications often use a higher wavelength (785 nm) compared to that of standard instruments to reduce fluorescence interference. A downside to this adaptation is the increased size of isotropic scatterers at this wavelength. Based on Wyatt’s conclusion, a 785 nm laser will scatter light equally in all directions for particles up to 26 nm [60]. Hence, the radius of gyration cannot be determined for particles under 26 nm. Gidh et al. used a laser of 690 nm and calculated the radius of gyration of unlyophilized lignin to be around 20 nm [53]. The particle size determined by Gidh et al. is about 1/35 of the laser’s wavelength [53]. This would put the ratio just beyond the limit established by Wyatt but also well below that of the smallest wavelength-to-particle ratio mentioned by Øgendal (1/60) [43,53,60]. As mentioned in the beginning of this section, the hypothetical particle-to-wavelength ratios listed here are based on the ideal spherical particle. The particle-to wavelength-ratio for lignin size determination may not match the ideal spherical guidelines.

To review: an opportunity to investigate the radius of gyration of lignin presents itself as its size lies near the border of isotropic and anisotropic scattering. This would allow complete derivation of calculating the absolute MW. Using common formalisms, the radius of gyration should be considered for future studies. Estimations of the scattering form factor can lead to precise MW calculations with the correction factor included in Equation (1). Electron or atomic force microscopy can provide supporting information on particle shape, in turn helping derive the radius of gyration [60]. Once a “library” of particle types (sphere, rod, coil) and their radii of gyration are obtained for different lignins in different solvent systems, then perhaps assumptions can be made for its respective form factor.

## 4. Second Virial Coefficient

A Zimm plot, shown in Figure 1, is used to determine the second virial coefficient, weight average MW, and radius of gyration [53]. As described previously, the second virial coefficient ($A_{2}$) has a small influence on the MW of common polymers when the approximation to the zero angle is used [41]. This procedure is briefly described below. As seen in Equation (1), $A_{2}$ is a part of a correction factor in relating MW to Rayleigh scattering. Outside of increasing accuracy of MW determination, $A_{2}$ is an important value that displays the interaction between a molecule and a solvent [59]. $A_{2}$ specifically relates the excess chemical potential between the molecule and solvent [58]. Simplifying Equation (1) with Zimm formalism (Equation (7)),
(7)$$\frac{K^{*}c}{R(\theta )}=\frac{1}{MP(\theta )}+2A_{2}c$$
allows one to determine the variables required for static light scattering [53,59]. The amount of scattered light at specific concentrations is projected to the zero angle or zero concentration to determine the radius of gyration and second virial coefficient, respectively [53]. Using the Zimm plot requires anisotropic scattering, otherwise there would be no angular dependence of scattering to extrapolate to the zero angle. Once both extrapolations are performed, the inverse of MW is defined by the y-intercept (Figure 1) [53]. Detailed calculations were articulated by Gidh et al. previously [53].

Gidh et al. applied the principles of the Zimm plot to determine lignin’s radius of gyration, second virial coefficient, and MW [53]. The physical relationships of the Zimm plot are illustrated by Gidh et al. (Figure 1), and it is the only study in this review to apply the Zimm plot to lignin [53]. A series of concentrations were tested in a MALS system, in which data were plotted for the scattered light recorded at each detector angle. Extrapolation of both terms to the zero angle and zero concentration created a y-intercept that was equal to the inverse of the MW. The slope of the linear regressions determined the radius of gyration and second virial coefficient for the zero angle and zero concentration extrapolations, respectively. More recent studies have not presented the Zimm plot and focused on MW determinations that do not require the identification of $A_{2}$ or radius of gyration values.

If the second virial coefficient is larger than zero, the solvent is considered “good” in terms of significant solute–solvent interactions. When this value is near zero, the solubility in the given solvent is low, and less than zero means a non-solvent [58]. The determination of $A_{2}$ will help with overall lignin characterization by putting a true value on solubility in different solvents. It is of note that, according to Equation (1), the second-term polynomial correction is more significant at larger MW and higher concentrations, being proportional to them. Estimating MW values from a MALS instrument may not require it, but with the demand of high lignin concentrations, knowing the $A_{2}$ value may be beneficial.

To the best of our knowledge, only one study has reported the determination of second virial coefficients for lignin samples [53]. Gidh et al. obtained a range of pH-dependent second virial coefficients [53]. For unlyophilized lignin, the Debye and Berry formalisms resulted in obtaining second virial coefficients of $1.535\times 10^{-3}$ and $3.335\times 10^{-3}$ mol mL g^−2^ from, respectively, from Zimm plots [53]. According to Striegel, a coefficient close to zero defines a poor solvent. At the same time, THF is known to be a good solvent for polystyrene with a second virial coefficient of $4.11\times 10^{-4}$ [58]. This would imply that solvents used by Gidh et al. are favorable for the tested lignin [53]. The solvent mixtures used by Ponnudurai et al. and future researchers will need to be investigated for effective solvation [1]. Each solvent inside the system may provide different effective solvation. Thus, lignin may favor one solvent over the other. Due to the wide variety of lignins and their limited solubility in common solvents [1], exploration of new solvents is bound to occur when applying SEC–MALS. Instrument stability, column compatibility, and lignin solubility all have to match fully. If a mixture of two solvents is to be applied, the determination of the second virial coefficients of each solvent is of utmost importance [58].

Striegel has shown how effective solvation can skew MW determination when solvent mixtures are used in SEC–MALS [58]. The error is due to analyte solvation that occurs unequally in the preferred solvent, causing the region surrounding the analyte to become enriched in the favored solvent [58]. Striegel has determined that differential detectors and light scattering detectors inaccurately portray the makeup of the solvents in the confined region around the analyte within a mixed solvent system [58]. This problem has been demonstrated by a solvent system containing two different second virial coefficients for SEC–MALS of polystyrene (18 kDa, 420 kDa, and 800 kDa) [58]. The wide range of polystyrene samples resulted in errors between several thousand and 10^5^ g mol^−1^ [58]. When using two solvents with nearly equal second virial coefficients, the same polystyrene samples reported statistically identical MWs as single-solvent systems [58].

In short, Striegel emphasizes the importance of the second virial coefficient when determining MW via SEC–MALS. Considering the impact of the second virial coefficient on the accuracy of MW values within mixed solvent systems, this value should not be ignored when two solvents are used. Lignin’s poor solubility increases the labor needed to obtain the second virial coefficient, but consideration should be taken when avoiding its determination in future studies. Simple Zimm plot analyses yield multiple sample characteristic values such as MW, radius of gyration, and second virial coefficients. Future lignin studies may benefit from the further characterization brought on by the Zimm plot.

## 5. Fluorescence Interference

The MW values for lignin determined in an early study with MALS were as high as $5.3\times 10^{5}$ g mol^−1^ (Table 1) [53]. Gidh et al. explained this finding as the aggregation of kraft lignin that decreases in size with an increase in pH [53]. However, another explanation can be the fluorescence behavior of kraft lignin [1], suggesting that abnormally high MW values were most likely due to an artifact of fluorescence interference. The intensity of scattered light is a fraction of the intensity of fluorescence. Therefore, fluorescence cannot be neglected. If a sample within a light scattering detector can fluoresce, it will lead to the misinterpretation of the amount of scattered light. In the case of MALS employed for MW determination, fluorescence emission will skew the theoretical relationship towards an overestimation of lignin’s MW (as seen in Table 1) [30].

The fluorescence of lignin is due to chromophores and conjugated structures, which can be further enhanced by aggregation within a solvent [30]. Recent studies employed several techniques to diminish fluorescence interference. The most common method in lignin studies is the use of fluorescence filters [7,30,54,55]. Ponnudurai et al. reported a forward monitor (recording possible absorbance by sample) change of only 0.15% for organosolv lignin [1]. This observation suggests that fluorescence mitigation is not needed when working with this low-fluorescence lignin [1]. Zinovyev et al. used an infrared laser, fluorescence filters, and a laser forward monitor to ensure that any absorption of light was accounted for, while minimizing fluorescence interference [7]. Gaugler et al. directly tested samples with and without fluorescence filters and observed large overestimations of molar mass due to fluorescence interference (Table 1) [54].

The studies referenced in Table 1 show how fluorescence mitigation changed the determined MW values. This conclusion is confirmed as technical lignins show lower MW values when fluorescence filters are used (Table 1). The use of fluorescence filters and infrared lasers, however, may not eliminate errors in MW determination [7]. Zinovyev et al. showed that MALS with fluorescence filtering predicted a size of 10^4^ g mol^−1^ at the end of a SEC elution profile of lignin. Comparing this value to those of lignin model compounds of similar elution times, the end of the chromatogram suggested a size of only several hundred g mol^−1^ [7]. This discrepancy pointed towards unfiltered fluorescence interference [7].

Another promising approach used by Zinovyev et al. was an absorbance correction to mitigate excess fluorescence interference (Figure 2a). The high MW species’ light scattering at the beginning of the chromatogram is concluded to have minor fluorescence interference [7]. The basis of this phenomenon stems from the fluorescence signal being dependent on concentration while scattering correlates to the product of concentration and molar mass. Therefore, in early eluting particles, the scattering dominates when compared to the later eluting particles, where higher concentration leads to increased fluorescence [7]. The laser monitor (LM) measures the intensity of incident laser energy, while the forward monitor (FM) measures light intensity after the flow cell [7]. Therefore, FM allows the detection of any light absorbance while in the flow cell due to the difference in FM and LM.

The profile of high MW species (Figure 2a, blue line) has been used to extrapolate the profile of lower MW species with FM, i.e., absorbance correction. Figure 2b displays the decrease in absorbance that occurs when using an infrared laser [7]. The extrapolation from high MW (Figure 2a) results in lower MW values than what was determined by the FM correction [7]. The non-linear response in the bulk region of the chromatogram (Figure 2a, blue dots) implies that some fluorescence interference still occurs, even with absorbance correction and fluorescence filters. It was concluded that this interference was due to excess fluorescence emission capable of passing between the narrow bandgap of the fluorescence filters [7].

Zinovyev et al. concluded that a portion of fluorescence passed through the bandwidth filters [7]. This led Pittman et al. to develop the transmission factor determining the amount of fluorescence emitted inside the bandgap filters [62]. The introduction of the transmission factor enabled the determination of accurate molar masses of fluorescent bovine serum albumin [62]. Determining the transmission through the bandgap filters ($T_{B}(\lambda )$) and the intensity ($I_{f}(\lambda )$) measured at each wavelength led to the generation of a transmission factor from Equation (8) [62].
(8)$$T_{f}=\frac{\int_{\lambda_{MALS}}^{\infty }I_{f}(\lambda )\cdot T_{B}(\lambda )}{\int_{\lambda_{MALS}}^{\infty }I_{f}(\lambda )}$$Any light intensity measured above the MALS laser wavelength ($\lambda_{MALS}$) is due to fluorescence [62]. Even though this development has not been applied to lignin, it was successful for natural proteins (for which the exact MW is known through sequencing). Pittman et al. demonstrated the applicability of the transmission factor by determining the MW of bovine serum albumin with a bound fluorophore. SEC–MALS analysis with the transmission factor results in MW values similar to the theoretical values [62].

In summary, the determination and subsequent application of transmission factors may lead to increased accuracy of MW values for fluorescent materials such as lignin. So far, limiting the fluorescence interference has greatly improved MALS accuracy, and eliminating interfering artifacts from the system via transmission factors is another step towards obtaining the true, absolute MW.

## 6. Coupling Fractionation with Multi-Angle Light Scattering

Fractionation methods are vital for accurate MALS measurements, as a monodisperse sample is required for interpretive formulas to hold true [60]. An unfractionated, polydisperse sample minimizes the shape-dependent data provided from light scattering [60]. The combination of SEC and MALS (SEC–MALS) instrumentation is the most common method for the analysis of lignin with MALS. This section highlights the differences between several separation methods: determining relative MW (SEC) and absolute MW (SEC–MALS, AF4–MALS). SEC relies on minimal interactions between the column stationary phase and the analyte. Due to the variable functionality, heteropolymeric, and branched structure of lignin, column interactions are almost guaranteed, resulting in longer elution times and incorrect molar mass determination [1,29,63]. The evaluation of column interactions is key to the successful application of SEC for lignin’s MW determination [29]. Increasing the ionic strength by adding salts to the mobile phase may also help limit non-steric interactions [63]. Recently, the successful calibration of a hydrophobic gel permeation stationary phase with both polar and non-polar standards has been demonstrated by Andrianova et al. [29]. Lignin’s interactions with different column packing materials have been observed and presumed to be due to the hydrogen bonding between aromatic materials and the stationary phase [29].

Several alterations to the standard SEC protocol have been tested to determine accurate lignin’s MW due to the shortcomings of representative analytical standards. Coupling a SEC column to a mass spectrometer with appropriate ionization techniques has been a recent trend in the analysis of biomass-derived products [63,64,65]. Nonetheless, the bias towards lower MW fragments should be taken into account since ionization and vaporization of larger MW species is hindered [38]. The benefit of MS detection is that characteristic ions of the analyte can be identified while providing a mass distribution [66].

A direct comparison of lignin SEC–MALS vs. lignin SEC analysis was performed by Ponnudurai et al. as shown in Figure 3a [1]. Polyethylene glycol and polyethylene oxide were used as standards for MW determination via SEC [1]. The relative MW of the feedstock lignin was determined to be 11,000 g mol^−1^, while the absolute MW was 12,600 g mol^−1^ [1]. Several acetone-dissolved lignin fractions were tested, and the deviation between SEC and SEC–MALS, seen in Figure 3a, was deemed significant [1]. It is apparent that the MW values determined by SEC are lower than those determined by SEC–MALS. Figure 3b shows a visual deviation of lignin’s MW from the standards via SEC–MALS [1]. It is of note that MALS determines only the weight-average MW [43], whereas the MW determined by SEC (including SEC–MALS) can be calculated as either number average or weight average [29,41]. Thus, for correct comparison with MALS, only the former should be evaluated. It has been made clear that without proper experimental methods and procedures, relative MW (SEC) will deviate from absolute MW (SEC–MALS) [1,29]. This difference suggests that lignin is likely branched but folded and more compact than the standards when in a solution [1]. This difference reflects the impact of unaccounted for column interactions, since SEC MW determination depends largely on the hydrodynamic radius and shape.

A less common fractionation technique of AF4 has increased usage in the field of light scattering. This is largely due to the lack of a stationary phase when compared to its SEC counterpart [67]. The lack of a stationary phase consequently reduces the unwanted interactions that often hinder SEC MW determination of complex materials [28,67]. These unwanted interactions are not nearly as impactful in absolute MW determination via SEC–MALS, but instead shedding the column stationary phase adds unwanted scatterers to the flow cell, skewing MW calculations [60]. Very limited application of AF4 has been attempted for lignin analysis largely due to the harsh chemicals needed for solubilization [28]. Dimethyl sulfoxide, a common lignin solvent, is yet to be compatible with the available AF4 membranes [67]. Aqueous-based lignin solvent systems have the ability to apply AF4 techniques with the careful selection of the molecular weight cut-off membrane and optimization of the cross-flows for sufficient separation [28].

Simply put, fractionation is crucial in determining the MW value from a MALS experiment. The theoretical relationship of MALS allows for more accurate MW determinations compared to that of its SEC counterpart, which relies on calibration standards. Further advancements in fractionation methods will enable more complex materials like lignin to be analyzed by MALS. Hence, higher-accuracy MW values of individual particles will be achieved instead of the combined particles’ average.

## 7. Conclusions and Future Directions

Based on a theoretical relationship between light scattering and molar mass, MALS has an advantage over other characterization methods by providing the absolute MW. An analyte that creates column interactions, such as lignin, is an ideal candidate for MW determination via MALS. However, the successful application of MALS for lignin’s MW determination requires the navigation of several key technical challenges. The major challenges include fluorescence interference, proper solubilization, and appropriate fractionation. Further lignin characterization may be achieved by determining the radius of gyration and second virial coefficient.

A clear understanding of the impact that fluorescence creates in a MALS system has been displayed in recent literature. Several successful studies address fluorescence mitigation through narrow bandgap filters. Another possible approach is the application of transmission factors to eliminate fluorescence artifacts that persist beyond the capabilities of fluorescence filters. Equation (1) has been applied only as its linear approximation in the published papers focusing on lignin’s MW. When parameters such as the second virial coefficient, radius of gyration, scattering form factor, and differential refractive index increments can be accurately calculated for lignin, the correction factor in Equation (1) can yield more accurate absolute MW values. The ever-important step of fractionation must be maintained to achieve monodisperse samples whose light scattering data can be related to size and MW. The further development of both fractionation methods of SEC and AF4, capable of withstanding polar organic solvents, could broaden the application of MALS to lignin analysis.

A standard or conventional method needs to be established within the field of lignin’s MW determination via MALS. The variation in methods and instrument setups creates minimal room for validation between laboratories. The standardized protocol, supported by recent studies, should include a MALS instrument with fluorescence filters coupled with an RI detector of matching wavelength. The method of differential refractive index increment determination needs to be considered with extreme care as it can develop large errors in MW. “Online” dn/dc approximations should be compared to the “batch” method to analyze any method discrimination in lignin samples. Lignin particle scattering needs to be verified as an anisotropic scatterer before reporting its radius of gyration. The use of common instrumentation and methods will be the onset of validated findings for lignin’s MW. Determining the second virial coefficient for lignin solvents may suggest a solvent of choice. Having a specified solvent will confine the spectrum of experimental parameters, leading to more comparable data in the future. Secondary confirmation methods of MW, independent of particle shape, should be applied for supplementary comparison to light scattering data. If possible, similar solvent systems should be tested for a range of lignins to create a better understanding of not only lignin but also how each experimental variable impacts the determined MW.

## Figures and Tables

**Figure 1 polymers-16-00477-f001:**
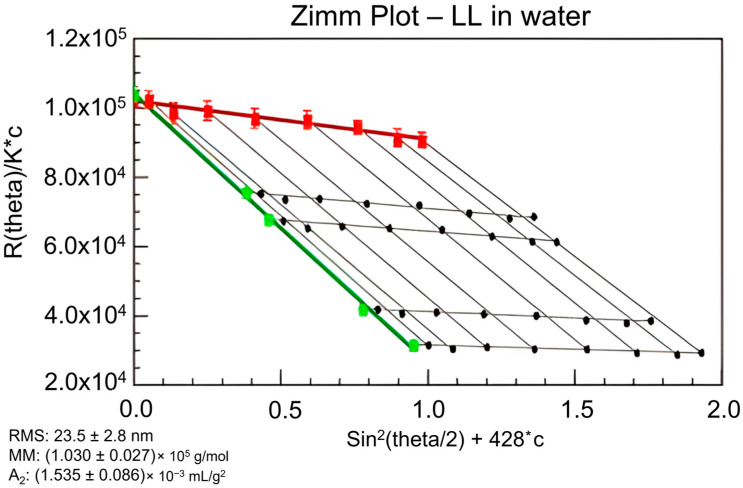
Zimm plot of unlyophilized lignin in water using Debye formalism. Black dots represent different concentrations of lignin measured at each angle. The green extrapolation is to zero concentration. The red extrapolation is to zero scattering angle. Reprinted from [53]. Copyright 2005, with permission from Elsevier.

**Figure 2 polymers-16-00477-f002:**
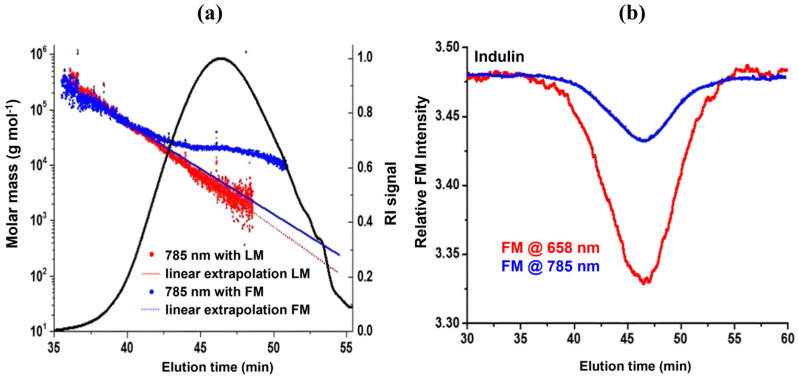
SEC–MALS analysis of kraft lignin (Indulin AT): (**a**) chromatogram displaying LM vs. FM extrapolation. (**b**) Forward monitoring (FM) displaying absorbance in flow-cell at two different wavelengths. Reprinted from [7].

**Figure 3 polymers-16-00477-f003:**
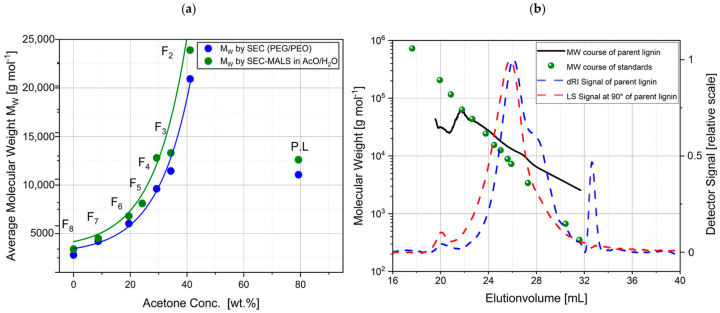
(**a**) Lignin fraction’s MW determined via SEC (blue) and SEC–MALS (green). (**b**) SEC–MALS MW comparison of PEG and PEO standards vs. feedstock. Reprinted from [1].

**Table 1 polymers-16-00477-t001:** Comparison of dn/dc and MW calculations based on procedural method and instrumentation for various technical lignins and their fractions.

Solvent	Laser λ (LS/RI)	Fluorescence Filters	Method (dn/dc)	Lignin Type	dn/dc (mL g^−1^)	MW (g mol^−1^)	Ref.
H_2_O	690/690 nm	N/A ^a^	Batch	Kraft	0.1928	540,700	[53]
0.1 M NaOH					0.1998	507,600	
DMF/LiBr	637/950 nm	N/A ^a^	Batch	Indulin AT	0.1340 *	319,000	[54]
				Organosolv Alcell	0.1330 *	827,000	
DMF/LiBr	637/950 nm	13 nm bandwidth	Batch	Indulin AT	0.1340 *	44,600	[54]
				Organosolv Alcell	0.1330 *	65,600	
THF	785/785 nm	10 nm bandwidth	Batch	Acetylated Indulin AT	0.0908	74,470	[30]
				Acetone insoluble	0.0823	82,470	
				Acetone soluble	0.1003	18,830	
Acetone/H_2_O	664/NR ^b^ nm	N/A ^a^	Batch	Organosolv Fabiola (Feedstock)	0.1865	12,600	[1]
				F_1_ (80 wt % acetone)	0.1836	13,000	
				F_2_ (60 wt % acetone)	0.2131	23,900	
				F_3_ (60–40 wt % acetone)	0.2097	13,300	
				F_4_ (40–35 wt % acetone)	0.1826	12,800	
				F_5_ (30 wt % acetone)	0.2219	8100	
				F_6_ (25 wt % acetone)	0.2168	6800	
				F_7_ (20 wt % acetone)	0.2135	4500	
				F_8_ (10 wt % acetone)	0.1632	3400	
DMSO/LiBr	785/633 nm	10 nm bandwidth/absorbance correction	Online	Indulin AT	0.1515	14,000	[7]
				Organosolv Alcell	0.1666	27,000	
				Lignoboost	0.1550	16,000	
				Soda sarkanda	0.1503	19,000	
				Sodium lignosulfonate	0.1187	55,000	
DMSO/LiBr	785/660 nm	10 nm bandwidth	Online	Hardwood lignosulfonate	0.120	15,080	[56]
				Softwood lignosulfonate	0.110	45,570	
				Hardwood kraft	0.150	4110	
				Indulin AT (softwood kraft)	0.160	13,950	

^a^ N/A denotes “not applicable” as filters were not used; ^b^ NR denotes “not reported”; * denotes reused value.

## Data Availability

Not applicable.

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
