# Peer review of "Advancing Molecular Weight Determination of Lignin by Multi-Angle Light Scattering"

_polymers, 2024, doi:10.3390/polym16040477_

Round 1

Reviewer 1 Report

Comments and Suggestions for Authors

This review article entitled “Advancing Molecular Weight Determination of Lignin by Multi-Angle Light Scattering” describes the molecular weight determination of lignin by using multi angle light scattering. The topic of the review is interesting. However, there are some comments, which can enhance quality of manuscript:

1.      Some places grammatical and typographical errors are present in the manuscript. Please rectify the errors. There were several places in this manuscript where the word choices did actively impede understanding. Few places scientific terminology is missing in manuscript. For instance in abstract section line number 17 “Provided will be an overview of key challenges that affect accurate….” and many others.

2.      Introduction lacks the state of art challenges in MADLS and what is main importance of the current review article in this context. Improve the quality of figures.

3.      Critical analysis is missing for some of the sections. I believe there should be concluding remarks at end of each section highlighting limitations/benefits of studies.

4.      It will also be really interesting if author can tabulate the results for molecular weight of lignin available in literature calculated using MADLS alongwith the some of feature of instrument such as type of laser.  

Reviewer 2 Report

Comments and Suggestions for Authors

The effect of radius of gyration on the determination of accurate molar mass is markedly overestimated. To get the true molar mass the particle scattering function need not be known. The effect of particle scattering function is eliminated by the extrapolation of scattered light intensities to zero angle. Remember, the particle scattering function at zero angle equals unity.

Similarly, the second term of basic light scattering equation is eliminated by extrapolation to zero concentration. And in SEC online mode the concentrations of eluting molecules are that low that the second term of light scattering equation can be neglected without introducing significant errors in molar mass. See for example Table 4.4 of ISBN:9780470877975.

Figure 1 is just a schematic representation and not an example of the real data. I suggest showing a Zimm plot for a typical lignin sample.

Round 2

Reviewer 2 Report

Comments and Suggestions for Authors

The text is markedly improved compared to the previous version.